# DiffuSeq-v2: Bridging Discrete and Continuous Text Spaces for Accelerated Seq2Seq Diffusion Models

**Shansan Gong**[1]  **Mukai Li**[1]  **Jiangtao Feng**[2]  **Zhiyong Wu**[3]  **Lingpeng Kong**[1]

[1]The University of Hong Kong  [2]Independent Researcher  [3]Shanghai AI Lab

sansa933@connect.hku.hk jiangtaofeng0906@gmail.com lpk@cs.hku.hk

## Abstract

Diffusion models have gained prominence in generating high-quality sequences of text. Nevertheless, current approaches predominantly represent discrete text within a continuous diffusion space, which incurs substantial computational overhead during training and results in slower sampling speeds. In this paper, we introduce a soft absorbing state that facilitates the diffusion model in learning to reconstruct discrete mutations based on the underlying Gaussian space, thereby enhancing its capacity to recover conditional signals. During the sampling phase, we employ state-of-the-art ODE solvers within the continuous space to expedite the sampling process. Comprehensive experimental evaluations reveal that our proposed method effectively accelerates the training convergence by 4x and generates samples of similar quality 800x faster, rendering it significantly closer to practical application. [1]

## 1 Introduction

After diffusion models gained significant attention in the vision domain (Ho et al., 2020), progress has been made in applying them to text generation tasks, including constrained text generation in Diffusion-LM (Li et al., 2022) and sequence-to-sequence (Seq2Seq) text generation in DiffuSeq (Gong et al., 2023). Numerous subsequent studies demonstrate that diffusion models have achieved results comparable to traditional autoregressive models and non-autoregressive models in tasks such as machine translation (Yuan et al., 2022; Gao et al., 2022; Zheng et al., 2023) and summarization (Lin et al., 2022; Zhang et al., 2023; Mahabadi et al., 2023; Zhou et al., 2023). However, most of these works suffer from slow convergence during training and slow generation speed, particularly considering that these approaches require

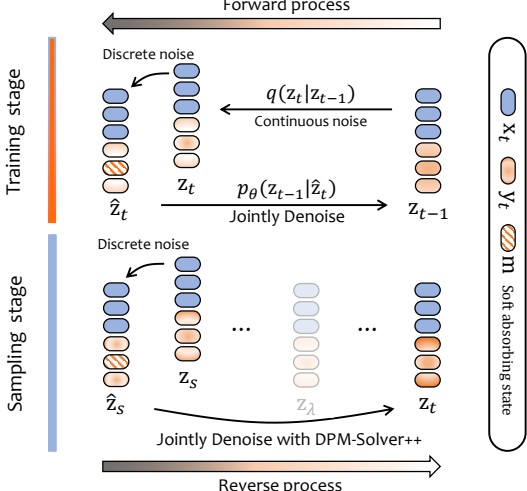

Figure 1: Training and sampling stages with discrete noise, which helps the two stages align better.

the Minimum Bayes Risk (MBR) decoding strategy (Koehn, 2004) to enhance generation quality, resulting in a doubling of time consumption.

In order to further narrow the gap between diffusion models and the prevailing autoregressive models, we aim to propose an accelerated version of DiffuSeq for both the training and sampling stages. For training, in addition to GPU acceleration techniques such as FP16, an improved training scheme can enable the model to better represent knowledge and learn data distribution more quickly (Hang et al., 2023). For sampling, a well-trained diffusion model can achieve similar quality within a single sampling and without the need for MBR decoding, thus saving generation time. Furthermore, we can borrow the state-of-the-art ODE sampler DPM-solver++ (Lu et al., 2022a,b) which is already applied in fast image generation. The progress in discrete text diffusion models (Zheng et al., 2023) exhibits its superiority in using fewer sampling steps, thus intuitively inspiring us to bridge the gap between continuous and discrete spaces.

Based on the BERT (Zhang et al., 2019) and

---

[1]The code is released at https://github.com/Shark-NLP/DiffuSeq/tree/diffuseq-v2.

BART (Lewis et al., 2020), as well as the absorbing state in D3PM (Austin et al., 2021), we propose incorporating an extra learned soft absorbing state and discretely adding it with Gaussian noise to jointly denoise the noise from two sources. The processes are illustrated in Figure 1. Specifically, after posing Gaussian noise, we randomly replace the continuous vector of the sequence with the absorbing state. The ratio is set according to the time step. This approach bridges the gap between continuous and discrete diffusion processes and also makes the training and inference stages better aligned. It also facilitates the integration of the DPM-solver++ and reduces the number of required sampling steps. In summary, our contributions are:

1. We introduce the learned soft absorbing state to help continuous diffusion models converge faster and eliminate the need for MBR decoding to ensure quality during sampling.

2. We adapt the DPM-solver++ to our enhanced diffusion text generation approach and demonstrate its feasibility in accelerating the generation speed experimentally.

## 2 Preliminaries

### 2.1 Continuous Diffusion Models

Ho et al. (2020) and Song et al. (2020) formulate diffusion models in continuous space including forward and reverse processes. The forward process gradually corrupts data point $\mathbf{x}_0$ into a standard Gaussian noise $\mathbf{x}_T \sim \mathcal{N}(0, \mathbf{I})$. For each forward step $t \in [1, 2, ..., T]$, the perturbation is followed by $q(\mathbf{x}_t|\mathbf{x}_{t-1}) = \mathcal{N}(\mathbf{x}_t; \sqrt{1 - \beta_t}\mathbf{x}_{t-1}, \beta_t\mathbf{I})$, with $\beta_t \in (0, 1)$ as different scales. After the forward process, the reverse denoising process tries to gradually reconstruct the original data $\mathbf{x}_0$ via sampling from $\mathbf{x}_T$ by learning a diffusion model $f_\theta(\mathbf{x}_t, t)$. Diffusion-LM (Li et al., 2022) and DiffuSeq (Gong et al., 2023) design an embedding function $\text{EMB}(\mathbf{w})$ to map the discrete text $\mathbf{w}$ into a continuous space and operate clamping on $\mathbf{x}_t$ to map it back to word embedding space at each sampling step to reduce rounding errors.

### 2.2 Discrete Diffusion Models

For discrete diffusion probabilistic models, each $\mathbf{x}_t$ is a discrete random variable as one-hot vectors in $\{0, 1\}^K$, indicating the current state of each token, where $K$ is the size of the vocabulary. Multinomial diffusion (Ho et al., 2020) adopts a uniform noise

distribution over the vocabulary. D3PM (Austin et al., 2021) specifies $q(\mathbf{x}_t|\mathbf{x}_{t-1})$ through a transition matrix, and makes it to be a point mass with the probability on an absorbing state [MASK]. Zheng et al. (2023) further derive an equivalent reparameterization to the discrete diffusion process. The resulting formulation is more amenable to training and leads to much-improved generation quality. However, discrete diffusion models may miss the opportunity to directly leverage the existing techniques from continuous diffusion models.

## 3 Methods

In DiffuSeq, it formulates Seq2Seq tasks as conditional generation and learns $p(\mathbf{w}^x|\mathbf{w}^y)$, where $\mathbf{w}^x$ and $\mathbf{w}^y$ refer to the input and target sequence separately. We follow the notation, concatenating two sequences as $\mathbf{z}_t = \mathbf{x}_t \oplus \mathbf{y}_t$, where $\mathbf{x}_t$ and $\mathbf{y}_t$ represent parts of $\mathbf{z}_t$ that belong to $\mathbf{w}^x$ and $\mathbf{w}^y$, respectively. To accelerate continuous diffusion models as well as leverage the absorbing state of discrete diffusion models, we add learnable soft absorbing state into DiffuSeq. We first combine the continuous Gaussian noise and discrete absorbing noise, and then jointly denoise them. The detailed derivations can be found in Appendix A.

### 3.1 Training Stage

**Forward process with soft absorbing state** In continuous space, we first add Gaussian noise $\epsilon$ for each time step with $\alpha_t = 1 - \beta_t$, $\bar{\alpha}_t = \prod_{i=1}^{t} \alpha_i$:

$$\mathbf{z}_t = \sqrt{\bar{\alpha}_t}\mathbf{z}_0 + \sqrt{1 - \bar{\alpha}_t}\epsilon. \quad (1)$$

Considering the $i$-th token in the hidden representation $\mathbf{z}_t$ of the word sequence, we replace its representation with our soft absorbing state $\mathbf{m}$ at a certain probability. The soft absorbing state $\mathbf{m}$ is in the same hidden dimension as word embeddings and is also jointly learned along with the whole diffusion process.

$$\hat{\mathbf{z}}_t^i = \begin{cases} \mathbf{m} & \text{if } \rho = 1 \\ \mathbf{z}_t^i & \text{if } \rho = 0 \end{cases}, \quad (2)$$

where $\rho = \text{Bernoulli}(\beta_t * \gamma)$, and $\gamma$ is the [MASK] ratio when $t = T$. This operation keeps the diffusion model in continuous space but discretely replaces the representation of some tokens in the sequence, and the replacement probability is also scaled to the time step, same with $\beta_t$. Noted that these two kinds of noise are posed partially on the target $\mathbf{y}_t$ space in the manner of DiffuSeq.

Table 1: Sequence-to-sequence text generation results on QQP. All results are reported without MBR decoding if not specified. The best result is bolded, and the gray columns are excluded from comparison considering the fairness. The sampling step of BG-DiffuSeq is 20. The relative improvement ↑ is computed between our speedup version (step=2) and the DiffuSeq (MBR=1).

| | Continuous space | | | Discrete Space (step=10) | | | | Mixed space (ours) | | | |
| | DiffuSeq (MBR=1) | DiffuSeq (MBR=10) | BG-DiffuSeq | Multi-nomial | RDM+ Multinomial | D3PM-absorbing | RDM+ absorbing | Original (step=2000) | Speedup (step=10) | Speedup (step=2) | ↑ |
|---|---|---|---|---|---|---|---|---|---|---|---|
| BLEU | 0.1884 | 0.2413 | 0.2077 | 0.2025 | 0.2315 | **0.2245** | 0.2313 | 0.2411 | 0.2210 | 0.2115 | 12.3% |
| R-L | 0.5327 | 0.5880 | 0.5652 | 0.5516 | 0.5712 | 0.5677 | 0.5735 | 0.5930 | **0.5732** | 0.5651 | 6.1% |
| BertScore | 0.7965 | 0.8365 | 0.8057 | 0.7966 | 0.8375 | **0.8281** | 0.8416 | 0.8393 | 0.8207 | 0.8036 | 0.8% |
| Speed (it/s) | 0.51 | 0.05 | 60.3 | 106.4 | 108.6 | 116.1 | 121.0 | 1.07 | 208.3 | **406.5** | $\approx 800\times$ |

**Jointly denoise** The reverse process is to jointly reconstruct the corrupted data point. The simplified loss function is almost the same with DiffuSeq, except for $\mathbf{z}_t$ in different noise strategies:

$$\min_\theta \mathcal{L}_{\text{VLB}} = \min_\theta \left[ \sum_{t=2}^{T} ||\mathbf{y}_0 - \tilde{f}_\theta(\hat{\mathbf{z}}_t, t)||^2 + \right.$$
$$\left. ||\text{EMB}(\mathbf{w}^y) - \tilde{f}_\theta(\hat{\mathbf{z}}_1, 1)||^2 + \mathcal{R}(||\mathbf{z}_0||^2) \right]. \quad (3)$$

### 3.2 Sampling Stage

Previous continuous text diffusion models adopt clamp operation to make the vector predictions more precise and reduce rounding errors (Li et al., 2022) during sampling. However, this operation is not deployed in the training stage, and the gap between training and sampling (Tang et al., 2023) may hinder the performance and the further optimization for sampling speed.

On the contrary, for our methods, during sampling, the same discrete noise in Eq. (2) is sprinkled in the continuous Gaussian noise, which bridges the training and inference in discrete space. Using the exact solution of diffusion ODEs proposed by DPM-solver++ (Lu et al., 2022a,b), given an initial value $\mathbf{z}_s$ at time $s > 0$, we have:

$$\mathbf{z}_t = \frac{\sigma_t}{\sigma_s} \mathbf{z}_s + \sigma_t \int_{\lambda_s}^{\lambda_t} e^\lambda f_\theta(\hat{\mathbf{z}}_\lambda, \lambda) \, d\lambda, \quad (4)$$

where the $\lambda$ is a strictly decreasing function of $t$, $\sigma_t$ is monotonic to $\beta_t$, and $f_\theta(\hat{\mathbf{z}}_\lambda, \lambda)$ is aligned with training objectives. The integral term can be analytically computed by repeatedly applying integration-by-parts $n$ times, and we can just approximate the first several orders and drop the high-order error terms. We use second order in the experiment.

## 4 Experiments

We try to validate two main research questions. RQ1: Can the soft absorbing state added in continuous diffusion models improve the generation

quality and boost the training convergence? RQ2: To what extent does the DPM ODE solver improve sampling speed and affect the performance?

### 4.1 Experiment setup

**Dataset** We adopt QQP [2] for paraphrasing the sentence to the same semantic content, which is lightweight to train and has been widely used in many Seq2Seq text diffusion models.

**Baselines** We choose DiffuSeq (Gong et al., 2023), BG-DiffuSeq (Tang et al., 2023) as the representative of continuous diffusion models. The latter is the enhanced version of DiffuSeq, targeting bridging the gap between training and sampling to generate high-quality texts within fewer steps. We choose multinomial (Hoogeboom et al., 2021), D3PM-absorbing (Austin et al., 2021), and a reparameterized version of them (Zheng et al., 2023) as the representative of discrete diffusion models, with fewer diffusion steps for training.

**Implementation details** We follow the training, sampling, and evaluation implementation of DiffuSeq[3]. The experiment is deployed on NVIDIA A100 80G GPUs, with 2 GPUs for training. More details can be seen in Appendix C.2.

### 4.2 Main results

As seen from Table 1, original DiffuSeq relies on the MBR decoding to ensure generation quality. By contrast, our method in the original sampling version achieves comparable performance without applying MBR by sampling 10 times, which significantly saves the time for generating high-quality texts. Even the speedup version (step=2) still outperforms DiffuSeq (MBR=1) a lot, according to the

[2] https://www.kaggle.com/c/quora-question-pairs
[3] https://github.com/Shark-NLP/DiffuSeq

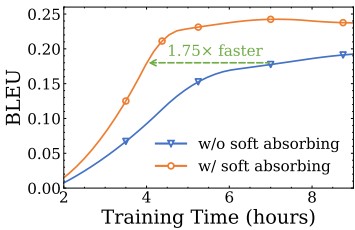
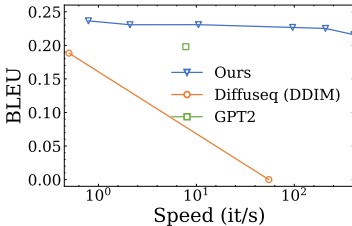
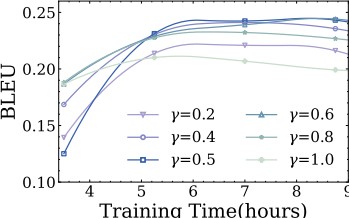

Figure 2: The test BLEU score along with training hours under different training schemes.

Figure 3: Generation speed and quality under different sampling steps incorporating DPM-solver.

Figure 4: The test BLEU score at different training hours for different settings of the ratio $\gamma$.

relative improvement. Our method is also superior to BG-DiffuSeq, which bridges the gap between training and sampling in continuous space. The speedup versions of our methods still have an advantage over discrete diffusion models, especially in terms of generation speed. We do not directly compare with RDMs because it designs an algorithm to route the discrete change of words, while ours just uses vanilla transition which uniformly changes tokens to the soft absorbing state.

### 4.3 Training speed

We use FP16 for GPU acceleration (Ott et al., 2018), which reduces the total training time from 28 to 11 hours with 2 GPUs and brings $2.5\times$ speed up. According to Figure 2, the jointly denoise training scheme expedites training convergence by at least $1.75\times$, probably because the absorbing state perturbs the representation of sequence discretely, which empowers the model with better capacity to reconstruct the discrete text information. The training consumption is saved more than $4\times$ in total.

### 4.4 Sampling speed

Sampling with FP16 is approximately $2\times$ faster than the original DiffuSeq. Furthermore, the incorporation of DPM-solver++ shrinks the sampling step to 10 or even 2 without sacrificing the performance much, as shown in Figure 3. This improvement is significant compared with DDIM (Nichol and Dhariwal, 2021) used in DiffuSeq. Comparing our speedup version (step=2) with DiffuSeq (MBR=1), texts with higher quality are generated conditionally and meanwhile about $800\times$ faster.

### 4.5 Ablation study

We test different sampling strategies in Table 2. After removing the clamp operation, the performance of DiffuSeq gets affected evidently, while ours seldom drop, which validates our previous assumption that adding the soft absorbing state bridges the gap

Table 2: Different sampling strategies. [C] denotes clamp operation and w/o [M] denotes stopping adding discrete noise during sampling.

|  | DiffuSeq | | Ours (Original) | | |
|---|---|---|---|---|---|
|  | w [C] | w/o [C] | w [C] | w/o [C] | w/o [M] |
| BLEU | 19.87 | 19.19 | 24.11 | 23.98 | 20.50 |
| Drop (%) | - | 3.4 | - | 0.5 | 14.9 |

between training and sampling, and we can remove the clamp operation. And further plugging in with DPM-solver++ will not introduce extra rounding errors. We test sampling without soft absorbing state on our model which still adds this discrete noise in the training stage. A significant drop can be seen, demonstrating the importance of the alignment of training and sampling. We also analyze the choice of the hyper-parameter $\gamma$. As seen in Figure 4, too small or too large [MASK] rate will harm the performance, and closer to the middle tends to perform better. So we choose $\gamma = 0.5$ as the default setting in our experiment.

## 5 Conclusions

In this work, we present a simple but effective training scheme for joint discrete and continuous text diffusion models, which additionally resets some tokens into the soft absorbing state. The discrete noise bridges the training and sampling stages, saving time consumption of these two stages and the plugged DPM-solver++ further makes the sampling faster. Our method is orthogonal to many other techniques such as self-conditioning (Mahabadi et al., 2023; Chen et al., 2022), choosing tokens with different importance (He et al., 2022), which can also further enhance generation quality. Our method is fundamental to diffusion text generation and can be applied beyond DiffuSeq (Chen et al., 2023).

## Limitations

Regarding the methods, we opt not to incorporate length prediction, unlike other approaches. Instead, we use the [PAD] token to indicate the length automatically. This may require more GPU memory for text generation. We validate our methods on the QQP dataset, which is one of the sequence-to-sequence text generation tasks. However, due to resource and time constraints, we are unable to test the effectiveness of our methods on more complex tasks such as machine translation and summarization. Additionally, this work does not explore the impact of scaling up the model size.

## Ethics Statement

In this study, we used diffusion models to generate text. To ensure that the generated text is free from bias, we carefully selected our training data and evaluated the quality of the generated text. We acknowledge that our research has potential ethical implications, particularly in the area of text generation for malicious purposes. To mitigate this risk, we have taken steps to ensure that our research is conducted in an ethical manner and that the generated text is used only for legitimate purposes.

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

## A  Detailed Derivations of Our Methods

### A.1  Training stage

Following Ho et al. (2020); Nichol and Dhariwal (2021); Song et al. (2020); Li et al. (2022); Gong et al. (2023), we define the forward noising process and reverse denoising process on the latent continuous space $\mathbf{z}$.

The *forward* noising is to perturb the structure of data $\mathbf{z}_0$. $\mathbf{z}_0$ is finally changed into the partial Gaussian noise with $\mathbf{y}_T \sim \mathcal{N}(0, \mathbf{I})$ through $T$-step forward random disturbance

$$q(\mathbf{z}_t|\mathbf{z}_{t-1}) = \mathcal{N}(\mathbf{z}_t; \sqrt{1-\beta_t}\mathbf{z}_{t-1}, \beta_t\mathbf{I}), \quad (5)$$

with $t = 1, 2, ..., T$ and $\{\beta_t \in (0,1)\}_{t=1}^T$ are the variance schedule. Let $\alpha_t = 1 - \beta_t$ and $\bar{\alpha}_t = \prod_{i=1}^t \alpha_i$, we have:

$$
\begin{aligned}
\mathbf{z}_t &= \sqrt{\alpha_t}\mathbf{z}_{t-1} + \sqrt{1-\alpha_t}\epsilon_{t-1} \\
&= \sqrt{\alpha_t\alpha_{t-1}}\mathbf{z}_{t-2} + \sqrt{1-\alpha_t\alpha_{t-1}}\bar{\epsilon}_{t-2} \quad (6) \\
&= ... = \sqrt{\bar{\alpha}_t}\mathbf{z}_0 + \sqrt{1-\bar{\alpha}_t}\epsilon,
\end{aligned}
$$

where $\epsilon$ stands for Gaussian noises. In the end, $q(\mathbf{z}_t|\mathbf{z}_0) = \mathcal{N}(\mathbf{z}_t; \sqrt{\bar{\alpha}_t}\mathbf{z}_0, (1-\bar{\alpha}_t)\mathbf{I})$. We use a sqrt noise schedule $\bar{\alpha}_t = 1 - \sqrt{t/T + s}$ with $s$ as a small constant at the start of the noise level.

Based on $q(\mathbf{z}_t|\mathbf{z}_{t-1})$, we further add discrete noise. Consider the $i$-th token in the hidden representation $\mathbf{z}_t$ of the word sequence, we replace its representation with our soft absorbing state $\mathbf{m}$ at a certain probability. The soft absorbing state $\mathbf{m}$ is in the same hidden dimension as word embeddings and is also jointly learned along with the whole diffusion process.

$$\hat{\mathbf{z}}_t^i = \begin{cases} \mathbf{m} & \text{if } \rho = 1 \\ \mathbf{z}_t^i & \text{if } \rho = 0 \end{cases}, \quad (7)$$

where $\rho = \texttt{Bernoulli}(\beta_t * \gamma)$, and $\gamma$ is the [MASK] ratio when $t = T$. The replacement probability $\rho$ is also scaled to the time step, proportional to $\beta_t$. All these two kinds of noise are posed partially in the manner of DiffuSeq, to make sure the condition signal $\mathbf{x}$ unchanged.

For *reverse* process, the continuous noise is accumulated as before, and defined as:

$$p_\theta(\mathbf{z}_{0:T}) := p(\mathbf{z}_T)\prod_{t=1}^T p_\theta(\mathbf{z}_{t-1}|\mathbf{z}_t), \quad (8)$$

$$p_\theta(\mathbf{z}_{t-1}|\mathbf{z}_t) = \mathcal{N}(\mathbf{z}_{t-1}; \mu_\theta(\mathbf{z}_t, t), \sigma_\theta(\mathbf{z}_t, t)), \quad (9)$$

where the $\mu_\theta(\cdot)$ and $\sigma_\theta(\cdot)$ is the predicted parameterization of the mean and standard variation of $q(\mathbf{z}_t|\mathbf{z}_{t-1})$ in forward process. Using Bayes' rule:

$$q(\mathbf{z}_{t-1}|\mathbf{z}_t, \mathbf{z}_0) = q(\mathbf{z}_t|\mathbf{z}_{t-1}, \mathbf{z}_0)\frac{q(\mathbf{z}_{t-1}|\mathbf{z}_0)}{q(\mathbf{z}_t|\mathbf{z}_0)}, \quad (10)$$

substitute Eq. (6) to it and we can get the parameterized mean of $q(\mathbf{z}_{t-1}|\mathbf{z}_t, \mathbf{z}_0)$:

$$\mu_t(\mathbf{z}_t, \mathbf{z}_0) = \frac{\sqrt{\alpha_t}(1-\bar{\alpha}_{t-1})}{1-\bar{\alpha}_t}\mathbf{z}_t + \frac{\sqrt{\bar{\alpha}_{t-1}}\beta_t}{1-\bar{\alpha}_t}\mathbf{z}_0, \quad (11)$$

and for brevity, we moit the coefficient of $\mathbf{z}_t$ and $\mathbf{z}_0$ as constants.

We can use the variational lower bound to optimize the negative log-likelihood $\mathbb{E}[-\log p_\theta(\mathbf{x}_0)] \leq \mathcal{L}_{\text{VLB}}$. The objective can be further rewritten to be a combination of several KL-divergence.

$$
\begin{aligned}
\mathcal{L}_{\text{VLB}} &= \mathcal{L}_T + \mathcal{L}_{T-1} + \cdots + \mathcal{L}_0 \\
&= \mathbb{E}_{q(\mathbf{z}_{1:T}|\mathbf{z}_0)}\Bigg[ \log \frac{q(\mathbf{z}_T|\mathbf{z}_0)}{p_\theta(\mathbf{z}_T)} \\
&\quad + \sum_{t=2}^{T} \log \frac{q(\mathbf{z}_{t-1}|\mathbf{z}_0, \mathbf{z}_t)}{p_\theta(\mathbf{z}_{t-1}|\mathbf{z}_t)} \\
&\quad + \log \frac{q_\phi(\mathbf{z}_0|\mathbf{w}^{x\oplus y})}{p_\theta(\mathbf{z}_0|\mathbf{z}_1)} - \log p_\theta(\mathbf{w}^{x\oplus y}|\mathbf{z}_0) \Bigg].
\end{aligned}
\tag{12}
$$

For $1 \leq t \leq T-1$, we compute the parameterization of $\mathcal{L}_t$ by substituting Eq. (11) to minimize the difference from $\mu_t$ and $\mu_\theta$ following Ho et al. (2020):

$$
\begin{aligned}
\mathcal{L}_t &= \mathbb{E}_{\mathbf{z}_0}\left[ \log \frac{q(\mathbf{z}_t|\mathbf{z}_0, \mathbf{z}_{t+1})}{p_\theta(\mathbf{z}_t|\mathbf{z}_{t+1})} \right] \\
&= \mathcal{C}\mathbb{E}_{\mathbf{z}_0}[||\mathbf{z}_0 - f_\theta(\mathbf{z}_t, t)||^2],
\end{aligned}
\tag{13}
$$

where $\mathcal{C}$ is a loss independent constant.

It is intuitively believed that the above continuous diffusion models learns aims to recover the corrupt data $\mathbf{z}_t$ with $f_\theta$ to $\mathbf{z}_0$. When we look back to our added discrete noise, since its directly added to the $q(\mathbf{z}_t|\mathbf{z}_0)$, which can be directly modeled by $f_\theta$. The $\mathcal{L}_t$ should be:

$$
\mathcal{L}_t = \mathcal{C}\mathbb{E}_{\mathbf{z}_0}[||\mathbf{z}_0 - f_\theta(\hat{\mathbf{z}}_t, t)||^2].
\tag{14}
$$

Then the optimization of training loss $\min_\theta \mathcal{L}_{\text{VLB}}$ can be further simplified as:

$$
\begin{aligned}
\min_\theta \mathcal{L}_{\text{VLB}} = \min_\theta \Bigg[ &\sum_{t=2}^{T} ||\mathbf{y}_0 - \tilde{f}_\theta(\hat{\mathbf{z}}_t, t)||^2 + \\
&||\text{EMB}(\mathbf{w}^y) - \tilde{f}_\theta(\hat{\mathbf{z}}_1, 1)||^2 + \mathcal{R}(||\mathbf{z}_0||^2) \Bigg].
\end{aligned}
\tag{15}
$$

### A.2 Sampling stage

DPM-solver++ (Lu et al., 2022a,b) is proposed totally based on the continuous diffusion models. According to it, we have the exact solution of diffusion ODEs, given an initial value $\mathbf{z}_s$ at time $s > 0$:

$$
\mathbf{z}_t = \frac{\sigma_t}{\sigma_s}\mathbf{z}_s + \sigma_t \int_{\lambda_s}^{\lambda_t} e^\lambda f_\theta \, d\lambda,
\tag{16}
$$

where the $\lambda$ is a strictly decreasing function of $t$, $\sigma_t$ is proportional to $\beta_t$, specifically, $\sigma_t = \sqrt{1 - \bar{\alpha}_t}$.

We need to approximate $\int e^\lambda f_\theta \, d\lambda$, which can be analytically computed by repeatedly applying $n$ times of integration-by-parts. According to the second order multistep DPM-Solver++ algorithm, the reconstruct of $\mathbf{z}_0$ relies on the $f_\theta$, after posing discrete denoise in our methods, the algorithm still applies since our $f_\theta(\hat{\mathbf{z}}_\lambda, \lambda)$ is exactly aligned with training objectives.

## B  Related Work

Continuous diffusion models are first applied in image generation (Song et al., 2020; Ho et al., 2020) and then applied in text generation (Li et al., 2022; Gong et al., 2023). Meanwhile, discrete diffusion models (Austin et al., 2021; Hoogeboom et al., 2021; Zheng et al., 2023) is designed for text generation. (He et al., 2022) and (Zhou et al., 2023) directly leverage the [MASK] token used in pretrained language models. By contrast, our method learns the soft absorbing state from scratch along with the whole diffusion process and bridges discrete diffusion with continuous space. The idea of absorbing state or discretely corrupt data can be seen in many NLP work like BERT (He et al., 2022) or BART (Lewis et al., 2020), or even in diffusion image generation (Hu et al., 2022).

## C  Implementation Details

### C.1  General setting

We use 2 A100 80G GPUs for training using FP16 with a batch size of 425 and a single GPU on sampling with a batch size of 100. The generation speed is measured under the setting of 100 batch size on one NVIDIA A100 80G GPU for all models, averaged by 3 runs.

To evaluate the quality, we use the standard metric BLEU (Papineni et al., 2002) and ROUGE (Lin, 2004) score. Since string-similarity-based metrics can be unsatisfactory for open-ended generation, we also report BERTScore (Zhang et al., 2019) that assesses the semantic similarity between generated sentences and references. For sentence-level diversity evaluation, we consider sentence-level self-BLEU (Zhu et al., 2018) to measure the n-gram overlap between the set of outputs w.r.t one source sentence. The self-BLEU score is computed using 2 samples for each test case generated with different seeds. All ablation study is conducted using the original version of our models if not specified, to validate the effectiveness of introducing the soft absorbing state.

## C.2 Baselines setting

All baselines we used are open-sourced. DiffuSeq[4] (Gong et al., 2023) and BG-DiffuSeq[5] (Tang et al., 2023) are implemented based on HuggingFace Transformers[6]. RDM[7] (Zheng et al., 2023) are implemented using Fairseq, where the temperature sampling is adopted when sampling and beam size is set to 1. For generation speed, we report the results on a single NVIDIA A100 80G GPU with batch size 100.

---

[4]https://github.com/Shark-NLP/DiffuSeq
[5]https://github.com/CODINNLG/Bridge_Gap_Diffusion
[6]https://github.com/huggingface/transformers
[7]https://github.com/hkunlp/reparam-discrete-diffusion