# OpenReview forum: "DiffuSeq-v2: Bridging Discrete and Continuous Text Spaces for Accelerated Seq2Seq Diffusion Models"
_EMNLP/2023/Conference — EMNLP 2023 Findings_

### Official Review · Reviewer_6MXf · 2023-08-06

**Soundness:** 3

**Excitement:**

3: Ambivalent: It has merits (e.g., it reports state-of-the-art results, the idea is nice), but there are key weaknesses (e.g., it describes incremental work), and it can significantly benefit from another round of revision. However, I won't object to accepting it if my co-reviewers champion it.

**Missing References:**

- A Cheaper and Better Diffusion Language Model with Soft-Masked Noise https://arxiv.org/abs/2304.04746 (contemporaneous work )

**Paper Topic And Main Contributions:**

The authors present a hybrid continuous/discrete diffusion model for text generation, which incorporates both the Gaussian noise of continuous diffusion models such as DiffuSeq, and the discrete word masking noise of discrete diffusion models such as D3PM.

The hybrid model is motivated by the desire to leverage the speed of the recently introduced DPM-solver++ for continuous diffusion models, and the goal of matching training to testing conditions by removing the need to "clamp" intermediate states to word embeddings at test time, which continuous diffusion models currently do.

Results on QQP demonstrate speedups over discrete diffusion models (DP3M and RDM) of about 2x, at similar performance levels.

**Questions For The Authors:**

- Does MBR further improve your method if the goal is performance? Since samples can be generated in parallel, I think this is important to touch upon.

**Reasons To Accept:**

- Well motivated. Hybrid diffusion models that combine the advantages of progress on continuous and discrete diffusion models are definitely worth exploring. The only paper on this I am aware of is so far unpublished (A Cheaper and Better Diffusion Language Model with Soft-Masked Noise https://arxiv.org/abs/2304.04746)

- Experiments seem to support the claims, and the effect of adding discrete masking on training and inference time, as well as performance, is adequately investigated (with caveats, see limitations section).

**Reasons To Reject:**

- The abstract focuses on comparisions to DiffuSeq (e.g. samples of similar quality 800x faster), which is deceptive. Discrete diffusion models are much faster, and almost as performant, as their results show. Their method is really a hybrid of the continuous DiffuSeq and the discrete D3PM, and should be discussed as such.

- Related, why DiffuSeq cannot use DPM-solver++ is not explicit enough. I assume that it is exactly because of incompatability with clamping (as Figure 3 suggests), but this should be clarified.

- In general, the paper is not in great shape, and was clearly prepared in a rush. Notation and acronyms are often not introduced, central ideas like "clamping" are not adequately summarized, and claims in the abstract (e.g. 4x faster training vs. 1.75) don't match those in the paper. Overall however, I could understand most of the paper adequately.

- Details around the experiments conducted are incomplete. For example, there is not enough information about the diffusion models (e.g. number of parameters etc.) to even judge if the comparisons are fair. Obviously this must be addressed before the paper can be accepted.

- All results are on a single dataset  (QQP). New DM approaches are generally evaluated on several tasks (but this is a short paper submission).

**Reproducibility:**

2: Would be hard pressed to reproduce the results. The contribution depends on data that are simply not available outside the author's institution or consortium; not enough details are provided.

**Reviewer Confidence:**

3: Pretty sure, but there's a chance I missed something. Although I have a good feel for this area in general, I did not carefully check the paper's details, e.g., the math, experimental design, or novelty.

**Typos Grammar Style And Presentation Improvements:**

- Figure 4 (lambda in figure, gamma in caption and text)
- clamping: describe operation(in footnote maybe?)
- w^y notation was not introduced
- "circle-plus" operator not introduced
- "training and inference stages aligned better'-> training and inference stages better aligned
- "these two kinds of noise are posed partially in 152 the manner of DiffuSeq." -> What does this mean?
- "loss function is almost the same with DiffuSeq:" -> complete the though, "except for ..."
- "applying n times of integration-by-parts"-> applying integration-by-parts n times

---

### Official Review · Reviewer_SUQ8 · 2023-08-09

**Soundness:** 4

**Excitement:**

3: Ambivalent: It has merits (e.g., it reports state-of-the-art results, the idea is nice), but there are key weaknesses (e.g., it describes incremental work), and it can significantly benefit from another round of revision. However, I won't object to accepting it if my co-reviewers champion it.

**Paper Topic And Main Contributions:**

This paper proposes a method to accelerate sequence-to-sequence text generation using diffusion models. The main contributions are:

a. Introducing a learned soft absorbing state that helps the continuous diffusion model converge faster during training and eliminates the need for MBR sampling to ensure quality.

b. Adapting  DPM-solver++ to expedite the sampling process of the enhanced diffusion text generation model.

The authors add discrete noise by randomly replacing tokens in the sequence with the soft absorbing state, in addition to the continuous Gaussian noise. This helps bridge the gap between training and inference. For sampling, DPM-solver++ allows accurate reconstruction using very few steps.

The method outperforms some baselines like Multinomial diffusion and DiffuSeq.

**Questions For The Authors:**

a. Is the methods you proposed generalizable? Can they be applied to various diffusion models, or can they only be used on Diffuseq?

b. Can you provide the experiments (BLEU, R-L, bertscore, generation speed) finetuning the t5/bart/vanilla-seq2seq-transformer using the same datasets?

**Reasons To Accept:**

This is a clearly written paper that makes useful contributions to accelerating the training and sampling of diffusion models for text generation. The method is simple but effective. The experiments comprehensively demonstrate accelerated training convergence, faster high-quality sampling, and strong performance compared to baselines.

**Reasons To Reject:**

a. It lacks detailed comparisons with more baselines. There are at least 10+ text-generation baselines using diffusion models. (https://github.com/diff-usion/Awesome-Diffusion-Models#natural-language)
So more experiments should be conducted to test the generalizability of the proposed techniques.

b. It lacks comparison with standard seq2seq Transformer text-generation baselines, such as a finetuned  T5 or BRAT. The final objective of using the diffusion model for text generation is to replace the traditional autoregressive/seq2seq model, so the comparison between diffusion and non-diffusion models is necessary.

c. The proposed techniques lack novelty. This paper just simply adds two methods based on the previous strong baselines. DPM-solver++ is very popular and common in computer-vision field and using it could not be viewed as a contribution.

**Reproducibility:**

3: Could reproduce the results with some difficulty. The settings of parameters are underspecified or subjectively determined; the training/evaluation data are not widely available.

**Reviewer Confidence:**

5: Positive that my evaluation is correct. I read the paper very carefully and I am very familiar with related work.

---

### Official Review · Reviewer_ZN1G · 2023-08-10

**Soundness:** 3

**Excitement:**

2: Mediocre: This paper makes marginal contributions (vs non-contemporaneous work), so I would rather not see it in the conference.

**Paper Topic And Main Contributions:**

The paper introduces a soft absorbing state to the continuous diffusion model to improve the generation quality and boost training convergence. The authors use an off-the-shelf continuous ODE solver during the sampling process to improve the sampling speed. The paper conducts experiments to validate the time cost of the training and sampling process while achieving comparable results to the baseline models.

**Questions For The Authors:**

1. The theoretical explanation or intuitive reason for introducing an absorbing state to the continuous diffusion models.

**Reasons To Accept:**

1. The proposed soft absorbing state method can boost the training convergence.

**Reasons To Reject:**

1. Adding soft absorbing state (mask strategy) to the continuous diffusion process is a rigid combination. The paper does not explain why introducing an absorbing state can lead to improvement.
2. Leveraging an ODE solver to accelerate the continuous sampling process is a straightforward method, which is too naive to be a novel contribution.
3. The improvement of generation quality is minor compared with D3PM, also worse than the DiffSeq with MBR and RDM, which weakens the effect of the proposed soft absorbing state method. The saved training time cost is minor as well.

**Reproducibility:**

3: Could reproduce the results with some difficulty. The settings of parameters are underspecified or subjectively determined; the training/evaluation data are not widely available.

**Reviewer Confidence:**

5: Positive that my evaluation is correct. I read the paper very carefully and I am very familiar with related work.

---

### Meta-Review · Area_Chair_Z6yT · 2023-09-16

**Recommendation:** 4

**Metareview:**

The paper proposes a modification for continuous diffusion models by introducing a soft absorbing state, which enables removing the clamping operation in models like diffusion-LM and DiffuSeq. This modification enabled the utilization for DPM-solver++ for speeding up continuous diffusion models applied in text. Experiments are provided for speed comparison to show significant speed-ups compared to other models based on continuous diffusion. The resulting speed-up in roughly equivalent to the speed of discrete diffusion models. The paper requires writing and presentation improvements to clarify some of the points raised by the reviewers.

---

### Decision · Program_Chairs · 2023-10-07

**Decision:**

Accept-Findings

**Comment:**

The paper proposes a modification for continuous diffusion models by introducing a soft absorbing state, which enables removing the clamping operation in models like diffusion-LM and DiffuSeq. This modification enabled the utilization for DPM-solver++ for speeding up continuous diffusion models applied in text. Experiments are provided for speed comparison to show significant speed-ups compared to other models based on continuous diffusion. The resulting speed-up in roughly equivalent to the speed of discrete diffusion models. The paper requires writing and presentation improvements to clarify some of the points raised by the reviewers.